# Characterisation of Bee Pollen from the Marche Region (Italy) According to the Botanical and Geographical Origin with Analysis of Antioxidant Activity and Colour, Using a Chemometric Approach

**DOI:** 10.3390/molecules27227996

**Published:** 2022-11-18

**Authors:** Sara Castiglioni, Mariassunta Stefano, Paola Astolfi, Michela Pisani, Patricia Carloni

**Affiliations:** 1Department of Agricultural, Food and Environmental Sciences—D3A, Università Politecnica delle Marche, Via Brecce Bianche, I-60131 Ancona, Italy; 2Amap Marche Agricoltura Pesca—Agenzia per l’Innovazione nel Settore Agroalimentare e della Pesca, Centro Agrochimico Regionale, I-60035 Jesi, Italy; 3Department of Materials, Environmental Sciences and Urban Planning—SIMAU, Università Politecnica delle Marche, Via Brecce Bianche, I-60131 Ancona, Italy

**Keywords:** bee pollen, antioxidant activity, polyphenols, colour, granulometry, spontaneous flora

## Abstract

Attempts have often been made to isolate and characterise monofloral pollens to correlate nutritional with botanical properties. Nevertheless, pollen harvested in a particular area that can have a high biodiversity could have healthier properties. In addition, the analysis of the pollen’s botanical composition can be important for characterising the typical flora of a specific geographical area. On this basis, various pollens collected in different locations of the Marche region (Italy) and in different harvesting periods were analyzed for botanical composition and antioxidant (total phenolic content, ABTS, DPPH and ORAC tests), granulometry and colour (CIE L*a*b*) properties to evaluate the biodiversity of pollen sources within a particular geographical area and to correlate this to the nutraceutical characteristics. Antioxidant activity results showed values generally higher than those of monofloral pollens harvested in the same areas but manually separated according to colour, shape and size. This suggests that even the floral species present in low percentages may have an influence on the nutraceutical properties of these products. The multivariate statistical elaboration of the obtained results permitted the separation of samples containing a prevalent botanical species and the grouping of all the samples into separate clusters corresponding to different areas of Marche.

## 1. Introduction

Pollen is the fertilizing element of flowers. It is contained in the stamens of flowers and is composed of very small granules that, depending on the flower of origin, can show different colours: yellow, green, pink, white, red and brown [1]. Passing from flower to flower, bees collect the pollen granules to produce their only protein food [2] and, at the same time, this process benefits pollination. Pollen then plays a fundamental role for the nutrition of the larvae, and since the quantity and nutritional quality of pollen influences the development of the brood and a diverse pollen diet is considered favorable for bee colonies, the knowledge of the composition and biodiversity of pollen sources within a geographical area is very important in the beekeeping sector [2]. In addition, the collection of pollen by bees plays a fundamental role in the conservation of plant biodiversity [3]. Furthermore, pollen is also a complete and precious food for humans at any age, since it contains many energetic, nutritional and therapeutic substances. It is part of the superfoods of biopharmaceuticals, and its consumption is continuously growing [1,4,5]. Considering that some of the bee pollen components are antioxidants with beneficial effects, the quantitative investigation of the antioxidant activity of bee pollen can in some way reflect the nutraceutical potential of these components [1,6].

In a previous study, the possibility of geographically classifying pollen from northern, central and southern Greece using the results of the palynological analysis of the samples was demonstrated, thus increasing the knowledge on the flora of Greece through the observation of the most widespread botanical families. In addition, Morais et al. in 2011 [7] studied the relationship between the geographical origin and the antioxidant activity of heteroflorous pollen samples coming from different Natural Parks of Portugal and found significant differences in the total polyphenol content and antioxidant activity between most of the samples taken in the different parks. These results led the authors to affirm that these differences are attributable to both geographical and botanical origin.

For this reason, attempts have often been made to isolate and characterise monofloral pollens to correlate nutritional with botanical properties [8]. Nevertheless, palynological analysis does not always provide insight on the pollen’s nutraceutical composition and potential therapeutic activities. The composition of pollen depends on the geographical as well as on the botanical origin, and mixes of pollen deriving from a particular area with high biodiversity could show healthier properties than monofloral pollen [9] and could affect bee physiology, helping us to better understand the influence of agriculture and land use intensification on bee nutrition [10]. In addition, studying the pollen’s botanical composition can also be important for characterising the typical flora of a specific geographical area [2].

On this basis, various pollens were collected in three different locations of the Marche Region (Italy) in different harvesting periods and analysed to evaluate variations in botanical composition over time and the biodiversity of pollen sources within a particular geographical area, as well as to correlate the antioxidant activity and other physical properties of the pollen samples with their botanical and geographical origin.

## 2. Results and Discussion

In this study, a total of 24 pollen samples belonging to three different apiaries located in different provinces of the Marche region were analyzed (Table 1) to identify the frequency of each pollen class in each flowering period and province and to correlate the botanical and geographical characteristic with antioxidant activity and chemical physical properties.

Data obtained from the palynological analysis crossed with floristic and phytogeographical studies concerning the predominant melliferous species spread in the studied region [11] show the dominant presence of *Fraxinus ornus* L., *Coriandrum* (Umbelliferae f. A), *Castanea* Mill., *Quercus ilex* gr. and *Trifolium alexandrinum* L. pollens together with a lower presence of *Olea f.*, *Vitis* L., *Prunus f.*, *Rubus f.* and *Salix* L. pollens.

### 2.1. Palynological Analysis

Palynological characteristics of the samples are reported in Figure 1, whereby percentages of each pollen type (predominant pollen, secondary pollen and important minor pollen up to 10%) are shown for each sample; the complete analysis is reported in the Appendix A. 

From the obtained data, it can be observed that 19 of the samples have a predominance of a pollen type greater than 50%, and that their composition depends both on the harvested season and location. The predominant species in the samples collected in the Pesaro province are the *Fraxinus ornus* L. (PU-FR01-05) in spring and the *Coriandrum* (Umbelliferae f. A) in summer (PU-CO06-09); samples harvested in the Ascoli Piceno province are instead prevalently composed of *Fraxinus ornus* L. (AP-FR02-03) in early spring and of *Quercus ilex* gr. (AP04-07) in late spring; during the summer, *Castanea* pollens (AP-CA08-12) are predominant. Samples coming from the Macerata province were harvested during a very short period and are all mainly composed of *Trifolium* pollen (MC-TR01-03). 

### 2.2. Total Phenolic Content and In Vitro Antioxidant Capacity 

Considering the different chemical structures of antioxidant compounds and the complexity and multiplicity of free radicals’ reaction mechanisms involved in the oxidative processes, to determine the in vitro antioxidant activity of a food matrix, it is usually necessary to use different methods and combine the obtained results [12]. In this study, the antioxidant potential of bee pollen extracts was determined by means of three different assays, namely ABTS (2,2′-azino-bis(3-ethylbenzothiazoline-6-sulfonic acid) radical cation-based), DPPH (2,2-diphenyl-1-picryl-hydrazyl-hydrate free radical) and ORAC (Oxygen Radical Absorbance Capacity) assays. In the ABTS test, the antioxidant capacity of the samples can be assessed using the reaction between the radical cation ABTS^•+^ and the antioxidant compounds; in the DPPH assay, a stable nitrogen-centred radical reacts with antioxidants by means of hydrogen/electron transfer mechanism, and finally, in the ORAC assay, the inhibition of the peroxyl radical oxidation is evaluated over time to provide a more reliable estimation of the antioxidant defense against oxidation stress when compared with the other available tests [13].

In addition, the Folin test (TPC) was used to determine the total phenolic content, and hence the phenolic antioxidants contained in the food, by means of a reducing agent. The results are reported in Table 2.

Antioxidant capacity results show a considerable variation between samples containing different botanical species and from samples containing a prevalence of the same species but coming from different locations (for example, TPC for PU-FR02 and APFR03), suggesting that both the botanical origin and the harvesting area could have an influence on the phenolic content and antioxidant properties of bee pollen samples. 

The results were analysed using Pearson’s correlation test in order to highlight significant correlations among the different analyses, obtaining fairly high and significant results in all cases except between the ORAC and DPPH test, in which the correlation was still significant but not very high (r = 0.556; *p* = 0.005). In order to have a simplified view of the data, the results were also mediated by province. 

#### 2.2.1. Total Phenolic Content (TPC Assay)

The TPC values obtained from the analysis of bee pollen samples (Table 2) show large differences, e.g., the GAE (gallic acid equivalents) has a value between 12.8 mg/g DW (dry weight) in MC-TR02 and 31.0 mg/g DW in AP-CA12 sample. Similar and low phenolic contents were obtained for all the MC samples mainly constituted of *Trifolium* pollen (mean: 13.2 mg/g DW), whereas AP samples containing mainly *Castanea* pollen gave considerably higher values (Mean: 21.3 mg/g DW). 

Overall, the results were comparable with the data available in the literature for bee pollen loads. In particular, Alimoglu and coauthors [9] reported TPC values between 15–27 mg GAE/g for different samples of monofloral and polyfloral bee pollens, and Ilie and coauthors [14] reported values from 11 to 16 mg GAE/g for bee pollen samples harvested from plant species of spontaneous flora during the spring in Romania. In addition, Gabriele and coauthors [15] reported TPC values of 24.8 (mg GAE/g) for *Castanea* bee pollen, and Salonen and coauthors [16] reported a lower phenolic compound content for *Trifolium* bee pollen.

#### 2.2.2. ABTS Assay

The results obtained using the ABTS assay are reported in Table 2, in which data are also grouped by province: similarly to the FOLIN results, the MC-TR02 sample showed the lowest value for TE (Trolox equivalents) (112 μmol TE/g DW), and AP-CA12 showed the highest value (258 μmol TE/g DW). Generally, all bee pollen samples coming from the apiary situated in the MC province had low values (112–128; mean: 120 μmol TE/g DW), whereas those obtained for samples from AP provinces were significantly (*p* < 0.05) higher (152–258; mean: 196 μmol TE/gDW).

#### 2.2.3. DPPH Assay

Antioxidant activity monitored with the DPPH assay showed values between 30.3 (AP-FR03) and 147.4 (AP-CA12) μmol TE/gDW (Table 2), and the mean values for bee pollen from the three provinces were not significantly different (mean for AP: 78.4, MC: 49.5; PU: 49.3 μmol TE/gDW). When the single samples are considered, the significantly highest (*p* < 0.05) values were those obtained from samples containing >75% *Castanea* pollen (AP-CA09-12). The AP-CA12 sample composed of 100% *Castanea* pollen, showed the highest value, also in this assay.

For ABTS and DPPH assays, the recorded values are difficult to compare with the results in the literature, mainly due to the different experimental conditions adopted [9,13,14,15].

#### 2.2.4. ORAC Assay

For the in vitro antioxidant capacity monitored by the ORAC assay, (Table 2), values varied from 300.1 to 801.6 μmol TE/gDW (Mean: 558.9 μmol TE/gDW) and are very close to those reported in the literature for Italian bee pollens (on average 534.3 μmol TE/g) [15] and for Brazilian bee pollens (133–576 μmol TE/g) [17].

### 2.3. Instrumental Colour Measurement

The pollen loads’ colour is a physicochemical parameter that plays a crucial role in characterising the samples. The colourimetric characteristics of the surface of pollen loads (I) and of the finely ground pollen samples (M) are reported in Table 3.

The results obtained with the CIE L*a*b* method show a high and consistent variability among the different pollen samples; in addition, the milled pollen samples (M) had slightly higher values for the coordinate L* compared to the corresponding integer loads (I). This may be due to the higher clarity of the central part of the load due to a minor light exposition: during the grinding process, the colour of the whole material becomes lighter. The observed differences in the colour between the whole pollen and the ground pollen loads could also be due to the multiflorality and to the non-homogeneity of the samples.

Moreover, the colour coordinates of pollen samples containing a higher percentage of the same predominant pollen were compared with loads of monofloral pollen harvested in the same region and previously analysed [8], leading to a good and significant correlation between the percentage of predominancy and the b* coordinate (data not shown). 

In addition, for these samples, the b* and a* values were comparable with the data reported in the literature [18]. The a* (red-green) and b* (yellow-blue) parameters of the pollen loads may be interpreted as a reliable index of the richness in pigments with antioxidant activity [19] and of a different mineral’s concentration related to the botanical origin [20].

### 2.4. Pollen Load Size Distribution

In Table 3, the granulometry of the pellets of the pollen samples is reported. Most of the pollen loads had a size between 2400 and 2000 µm in all samples; however, it can be observed that samples from the PU apiary contained a greater amount of smaller loads when compared with those coming from AP that were characterised by a higher percentage of loads in the range of 2800 and 2400 µm. Some of these AP samples are nearly monofloral, containing mainly *Castanea* pollen (AP-CA08-12), and the pollen size distribution we observe is similar to that reported in our previous study on monofloral pollen [19]. The remaining AP samples (AP01-07) had a similar pollen size distribution but with a higher percentage of big loads (>2400). The MC samples, which are mainly composed of *Trifolium* pollen, contained the smallest pollen loads, with a good percentage between 2400 µm and 1690 µm. This is in agreement with our previous results on monofloral *Trifolium* pollen loads, which were composed of a good percentage of loads with a diameter between 1400 µm and 2000 µm [19].

### 2.5. Protein and Moisture Content

The percentage (%) protein content of the pollen samples reported in Table 1 was relatively high in all the samples and varied considerably according to the geographical and the botanical composition of the samples (from 16% of AP-CA08 to 25% of AP-FR03). The results are in accordance with previous studies that report a good protein content in bee pollen [21].

In Table 1, the moisture content determined after the dehydration of the pollen samples is also reported and ranges from 10.5% to 22.2%. The heat treatment was performed at low temperatures, with the aim of avoiding nutrient loss and food spoilage.

### 2.6. General Consideration and Multivariate Analysis

Honeybee-collected pollen is usually a complex mixture of pollens from different botanical origins. However, the presence of a cultivated area near to the apiaries can provide the collected pollen with a monoflorality that could be associated with a nutraceutical relevance. For this reason, it is important to establish whether the prevalent presence in the loads of a particular type of pollen can be related to its antioxidant content. In fact, in a previous study, the loads of several bee pollen samples were manually separated according to colour, shape and size, allowing for the obtainment of 32 samples that were almost unifloral and were classified into 13 botanical families and analysed for antioxidant activity. 

In addition, the knowledge of the botanical origin of bee pollens collected in a certain area can provide an indication of the flora of that area, and this information can be used to characterise the provenience of the pollen. For instance, in a previous study concerning honey produced in the Marche region [11], the analysis of the chemical-physical, spectroscopic and antioxidant characteristics of the honey allowed for the characterisation of the origin area of that honey, also providing information on the characteristic flora of a given territory.

In this context, the elaboration of the results of the analysis of the antioxidant properties, size and colour of pollens collected in different locations in the Marche region gave us the opportunity to understand whether and how the antioxidant properties of these pollens could be correlated to the type of flowers visited by bees and to the zonal characteristics of the area in which the apiaries were placed. In fact, together with the prevailing flora, spontaneous blooms are also present in particular areas, and these could influence the quality of the apiculture practice [22] and then of the bee products. Some studies have already shown how the presence of a fraction of spontaneous flora in the vicinity of agricultural crops could enhance the agronomic productivity by improving the abilities of the bees [23].

Comparing the results obtained in the analysis of the antioxidant capacity of pollen samples reported in Table 2 with those previously obtained by our research group in the analysis of manually separated monofloral bee pollens [8], it can be seen that the antioxidant activity values measured are generally higher than those of monofloral pollens. Furthermore, if the data obtained from the pollen samples with a prevalence of a specific species are compared with those of monofloral pollens, the results are different for samples collected in different areas. This suggests that even the floral species present in low percentages may have an influence on the nutraceutical properties of these products. 

All these reasons led us to statistically investigate the results obtained from the analysis of these pollen samples with a botanical prevalence together with those obtained from monofloral pollens (manually separated according to colour, shape and size) of the same botanical origin.

PCA permits the extraction of systematic variations in a dataset and can be used for the classification of samples and interpretation of their differences and similarities. In this study, PCA elaboration performed on the antioxidant, colour and size data of pollen samples with the prevalence of species present both in unifloral [8] and actual samples (namely, *Castanea*, *Trifolium*, *Fraxinus* and *Coriandrum*) permitted for the separation of all the samples according to the prevalent botanical species without using the palinological information.

This PCA model that uses eleven variables led to two significant principal components (PC) with an eigenvalue > 1 that explained 75.6% of the total system variability. In Figure 2, the variance explained and the loading matrix for the first two principal components are reported. The first factor, PC1 (49.1%), includes the information deriving from antioxidant data, the percentage of big loads and the brightness of colour (L* variable); colour data (a* and b* variables) and the percentage of medium and smaller loads are instead mainly considered in PC2 (26.5%). 

The relative score plot of the first two factors is displayed in Figure 2 and shows that the samples of different prevalent botanical origins are well-differentiated, although some of the pollen samples overlap and fit into different groups. *Castanea* bee pollens characterised by a high polyphenolic content and a light colour are located in the right area of the graph, whereas *Fraxinum*, *Coriandrum* and *Trifolium* pollen are shifted to the left, in proportion to the brightness of their colour. The *y*-axis (PC2) displays the location of *Coriandrum* (higher a*) upward and *Castanea* (higher b*) downward.

To understand whether the palynological data and chemical-physical properties of the pollen samples can instead help to characterise the provenience of the pollen, a further statistical elaboration was made to find a simplified relationship of the samples with their area of production (Figure 3).

A hierarchical cluster analysis was performed to verify whether the data structure would be able to identify subgroups among the bee pollen samples. For the analysis, the whole dataset incorporating all the 24 bee pollen samples and 22 variables (palynological antioxidant and granulometry data: Appendix A, Table 2 and Table 3) was processed through the application of the dissimilarity ratio and the Ward algorithm using the Euclidean distance to space the cluster. The degree of dissimilarity of the samples is expressed in the dendrogram of Figure 3, which clearly suggests the formation of three main clusters, grouping together samples harvested in the same province.

The possibility of grouping all the samples into separate clusters corresponding to different areas of Marche indicates that the data on the antioxidant, palynological and granulometry properties of bee pollen contain useful information for the classification of the samples depending on their geographical origin.

In addition, the same data were submitted to PCA to establish whether this elaboration can also differentiate bee pollen samples with respect to their production sites. Applying PCA to the whole dataset (24 samples, 22 variables), it was possible to extract seven significant principal components (F) with an eigenvalue > 1 that explain 90% of the total system variability. The score plot of the first two components displayed in Figure 3 shows that the samples belonging to the different provinces are sufficiently differentiated along the first component, F1 (33.8%), that mainly includes the information obtained from the antioxidant and granulometry results and arranges samples with increasing antioxidant activity towards the right side of the graph. The second component, F2 (17.2%), utilises most of the palynological data to differentiate the samples.

## 3. Materials and Methods

### 3.1. Chemicals and Equipment

All chemicals were of the highest analytical grade. [2,2′-azinobis-(3-ethylbenzothiazoline-6-sulfonic acid) diammonium salt] (ABTS), 6-hydroxy-2,5,7,8-tetramethylchroman-2-carboxylic acid (Trolox), gallic acid (GA), Folin-Ciocalteu reagent (2N solution), potassium persulfate (K_2_S_2_O_8_), sodium carbonate (Na_2_CO_3_), 1,1-diphenyl-2-picrylhydrazyl (DPPH•), fluorescein (3′,6′-dihydrosyspiro[isobenzofuran-1[3H],9′[9H]-xanthen]-3-one) and [2,2′-azobis(2-methylpropionamidine) dihydrochloride] (AAPH) were purchased from Sigma-Aldrich Chemical Co. (Milan, Italy). Ultrapure water was generated from a Milli-Q system by Merck Millipore (Merck KGaA, Darmstadt, Germany) and was used for all the experiments. Methanol was UHPLC-grade from Merck. Spectrophotometric measurements were recorded on a microplate reader (Synergy HT, Biotek, Winooski, VT, USA).

### 3.2. Pollen Samples

Pollen samples were collected in 2015 in the Marche region, Central Italy, by professional beekeepers, from beehives equipped with bottom-fitted pollen traps, located in three different areas of the region: Isola del Piano (PU), Loc. Cavaceppo (AP) and Matelica (MC). In total, 24 bee pollen samples were collected using a scheduled cadence and in different periods to obtain a multiflorality representative of the flowering period of the plants. Beekeepers attested to the pollen’s geographical origin.

The pollen was cleaned of debris and kept in plastic bags at −21 °C until delivery to the laboratory, where the samples were dried at 35 °C for 3 days to reach a moisture content lower than 10%. Samples were kept in the dark at room temperature until analyses that were carried out within six to twelve months from harvesting.

The pollen samples described in Table 1 have been identified with an acronym of two letters indicating the province of origin (PU, AP, MC), two letters identifying the prevalent botanical species and a consecutive number.

### 3.3. Palynological Analysis

Pollen type identification [24] was performed using an optical microscope with total magnification (400× and 1000×), and the bee pollen samples were classified using different pollen morphology guides together with beekeepers’ indications.

Two grams of each pollen sample were vigorously stirred in 15 mL of water for 30 min. The suspension was further diluted with 45 mL of water and stirred again before pollen analysis.

Three small drops of the well-mixed pollen grain suspension were applied on a mi-croscope slide and dried on a heating plate, and a few drops of glycerin jelly were added before covering with the cover slide. Pollen grain counts were performed under the microscope. 

In some cases, the determination of the botanical species was not possible, since pollens of the different species belonging to the same genus had too many similarities. In these cases, only the genus (e.g., *Salix* L.) or the pollinic type (e.g., *Prunus* form) is reported. In other cases, the pollens of different genera belonging to the same family were very similar, and the botanical genus was determined by crossing palynological analysis with beekeepers’ indications. 

The results are expressed as a percentage of the pollen type in Figure 1, in which only species present in a percentage up to 10% are reported. The complete analysis is reported in the Appendix A. 

### 3.4. Preparation of Pollen Extracts

Each sample (0.5 g) of finely ground pollen was extracted by shaking with 5 mL of 70% aqueous methanol (*v*/*v*) for 5 min and centrifuged for 10 min at 8000× *g*. The super-natant was separated, and the solid residue was re-extracted. The extracts were pooled together and then stored at −20 °C until analysis. Each pollen sample was extracted in triplicate.

### 3.5. Determination of Total Phenolic Content (TPC)

Total phenolic content in the pollen extracts was determined using the Folin–Ciocalteu reagent [25]. Briefly, 50 μL of 7.5 % water-diluted pollen extract or of a 60 mM appropriately diluted gallic acid standard ethanolic solution (0–0.64 mM in water) were transferred into each well of a transparent 96-well microplate. Thereafter, 150 μL of a 10-fold diluted solution of the Folin–Ciocalteu reagent was added. The microplate was shaken and left to stand for 10 min in the dark. After this time, 100 μL of a 10% Na_2_CO_3_ water solution were added to each well. Samples were left to stand for 120 min at room temperature in the dark, and then absorbance was read at 760 nm against water as the blank. The results were expressed as mg gallic acid equivalents per g of dry pollen (mg GAE/g DW), using the linear regression gained from the gallic acid calibration curve.

### 3.6. Determination of In Vitro Antioxidant Capacity (ABTS, DPPH, ORAC)

The in vitro antioxidant capacity was evaluated by means of three different methods, namely, the ABTS, DPPH and ORAC assays. The results from each assay were expressed as μmol Trolox equivalents per g of dry pollen (μmol TE/gDW), using the linear regression obtained from a Trolox calibration curve.

The ABTS assay was carried out according to the method described by Re and co-authors [26]. In particular, the coloured radical cation (ABTS^•+^) was prepared by mixing a 7.0 mM aqueous ABTS solution with a 24.5 mM aqueous solution of K_2_S_2_O_8_ as the oxidizing agent in a 9:1 ratio and allowing the mixture to stand at room temperature in the dark for 12–16 h before use. The prepared ABTS^•+^ stock solution was then diluted ≈50-fold with water, to reach an absorbance of 0.9 ± 0.1 at 734 nm. For the assay, 30 μL of the 2.5 % water-diluted pollen extract, or of a 1.8 mM appropriately diluted Trolox standard ethanolic solution (0–0.30 mM in water), or water as a control were added into each well of a transparent 96-well microplate, followed by 270 μL of the diluted ABTS^•+^ solution. The microplate was shaken and left to stand for 120 min at room temperature in the dark; after this time, the absorbance of the solution was read at 734 nm against water as the blank. The antioxidant activity was determined as inhibition percentage using the following equation:% Inhibition A_734_ = (1 − A_s_/A_c_) × 100
where A_s_ is the absorbance at 734 nm of samples containing the pollen extract or standard; A_c_ is the absorbance of the control. 

For the DPPH assay, the method described by Prior and co-authors [27] was employed. Briefly, 100 μL of the 2.5 % water-diluted pollen extract, or of a 0.45 mM appropriately diluted Trolox standard ethanolic solution (0–0.15 mM in water), or water as a control were mixed with 200 μL of a 0.2 mM ethanolic DPPH• solution. After 15 min at room temperature in the dark, the absorbance of the solution at 517 nm was read against water as the blank on a transparent 96-well microplate.

The DPPH• scavenging activity was determined as inhibition percentage using the following equation:% Inhibition A_517_ = (1 − A_s_/A_c_) × 100
where A_s_ is the absorbance at 517 nm of samples containing the pollen extract or standard; A_c_ is the absorbance of the control. 

Lastly, the ORAC (oxygen radical absorbance capacity) assay was also used to measure the antioxidant capacity of the different pollen samples, as previously described [28]. Briefly, in each well of a solid black 96-well microplate, 25 µL of 0.20 % PBS (phosphate buffered saline 75 mM at pH 7.4) diluted pollen extract, of a 0.45 mM appropriately diluted Trolox standard ethanolic solution (4.5–95.0 mM in PBS), or PBS as a control were mixed with 150 µL of a 0.008 μM solution of fluorescein in PBS. After 30 min incubation in the dark at 37 °C, 75 μL of a 25 mM AAPH solution in PBS were rapidly added to each well, and fluorescence was recorded from the top every 120 s for 3 h, using an excitation wavelength of 485/20 nm and an emission filter of 528/20 nm. The kinetics showed a classic fluorescence decay due to fluorescein decomposition that was delayed in the presence of pollen samples or of Trolox standard solution. The AUC (area under the fluorescence decay curve) was automatically calculated by the analytical software Gen5 2.00.18 (Biotek, Winooski, VT, USA) connected to the Synergy HT reader. The net AUC for each standard/compound was obtained by subtracting the area of the control sample.

### 3.7. Instrumental Colour Measurement

The colours of the surface of pollen loads and of the finely milled pollen samples were determined using a Konica Minolta CR-400 (Konica Minolta, Sensing Inc., Osaka, Japan) chromameter equipped with a D65 illuminant and operating with CIE L*a*b* (L*: 0 to 100, a*: −green to +red, and b*: −blue to +yellow) colour space. Calibration was performed with the white-coloured calibration tile (Y = 86.6, x = 0.3188, y = 0.3364) prior to the measurements.

Approximately 3 g of each pollen sample were poured into a sample holder, and three readings were taken from each sample surface. The results of the colour coordinates are expressed as mean values from the three independent experiments (n = 3) and are reported in Table 3.

### 3.8. Pollen Load Size Distribution

Pollen load size distribution was measured by sieve analysis. Ten g of pollen were loaded into a series of six 20 cm-diameter sieve trays (from top to bottom: 2800, 2400, 2000, 1690, 1400 and 1000 µm hole diameters). After shaking the sample in the sieve trays for two minutes, particles retained on the sieves were collected and weighed. The weight of each solid fraction was compared to the weight of the total solid to obtain the mass percentage of solid held by each plate and to classify the pollen load into seven groups. Groups containing a minor percentage of pollen were put together to lead four groups reported in Table 3.

### 3.9. Total Protein Content (%) and Moisture Determination

Total nitrogen content was determined through the Dumas method (dry combustion method) [29]. Pollen samples were weighed (4.0 ± 0.1 mg) into small tin capsules and heated in a purified O_2_ stream to a temperature of 1000 ± 10 °C to promote the full oxidation of organic N. The analysis was performed using a CHNS-O Elemental Analyzer (EA 1110-CHNS-O, CE Instruments) equipped with an oxidation (chromium oxide)/reduction (pellets of pure copper) analytical column. The running time was set at 250 s, and acetanilide (C 71.09%; N 10.36%; H 6.71%; O 11.84%) was used as a standard molecule to calibrate the instrument. Factor 6.25 was used to convert the total nitrogen into proteins. Protein content was expressed as protein percentage (Table 1).

The moisture of the samples was determined after the drying, using the method used by ASSAM (Agenzia per i Servizi nel Settore Agroalimentare delle Marche). Two grams of pollen were ground in a mortar to obtain a homogeneous powder that was spread in a thin and homogeneous layer. The sample was inserted into a thermobalance and was progressively heated until it reached a temperature of 90 °C in 3 min, which was kept for 40 min. The thermobalance automatically calculates the weight loss. The results were expressed as moisture percentage and were used to calculate the protein content of the samples.

### 3.10. Data Analysis

The results of the TPC, ABTS, DPPH and ORAC tests were expressed as mean values from at least three independent experiments (n = 3), each performed in triplicate. Pollen samples were classified according to their province of origin, and the results were expressed as mean with standard deviation (SD) for the different samples and the different provinces.

Statistical differences were obtained through an analysis of variance (ANOVA), followed by Tukey’s multiple comparison test at a 95% confidence level (*p* ≤ 0.05). 

The results were also processed using multivariate chemometric techniques involving cluster analysis (CA) and principal component analysis (PCA) together with the data described in a previous study. All statistical treatments were performed using XLSTAT software (Addinsoft SARL, Paris, France).

## 4. Conclusions

In this study, several pollen samples belonging to different apiaries located in the Marche region were analyzed to identify the frequency of each pollen class in each flowering period and province. Moreover, the botanical and geographical characteristics were correlated with antioxidant activity and chemical physical properties.

Data obtained from the palynological analysis show the dominant presence of *Fraxinus ornus* L., *Coriandrum* (Umbelliferae f. A), *Castanea* Mill., *Quercus* ilex gr. and *Trifolium alexandrinum* L. pollens. The TPC (12.8 to 31.0 mg GAE /gDW) and antioxidants (ORAC: 300 to 802 μmol TE/gDW) results show large differences between samples and are comparable with the data available in the literature on bee pollen.

A comparison of these results with those previously obtained from monofloral bee-pollens shows higher values for mixed pollen, suggesting that the floral species present even in low percentages should have an influence on the nutraceutical properties of these products. Pollen samples with a prevalence of *Castanea*, *Trifolium*, *Fraxinus* or *Coriandrum* were separated according to the prevalent botanical species, with statistical elaboration of antioxidant, colour and size data, whereas the correlation of palynological and chemical-physical properties of the pollen samples shows the possibility to group all the samples into separate clusters corresponding to different areas of Marche.

In conclusion, the characterisation of antioxidant activity, colour and granulometry of pollen samples from the Marche region could be used to promote the production and the commercialisation of this bee product with high nutraceutical properties and to promote the diffusion of spontaneous species near productive landscapes to preserve the survival of bee colonies.

## Figures and Tables

**Figure 1 molecules-27-07996-f001:**
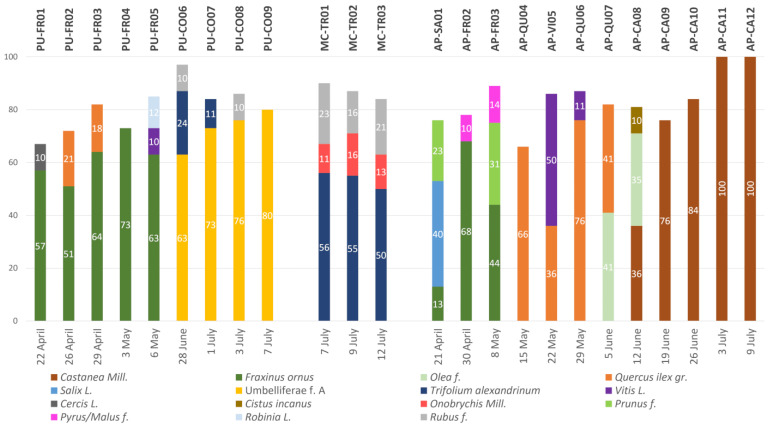
Palynological characteristics of bee pollen samples. Identification of pollen type and percentage (prevalence > 10 %), of harvesting zone (code) and period for each pollen sample are provided. The botanical species was also determined using the knowledge on the territory’s flora, the cultivation maps and beekeepers’ indications.

**Figure 2 molecules-27-07996-f002:**
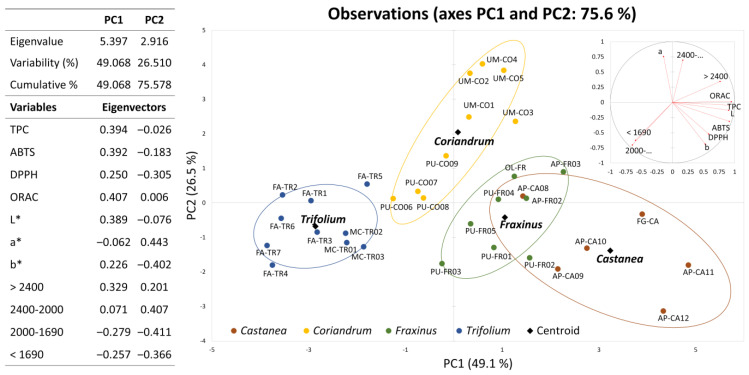
Principal component analysis. Loading plot and score plot of the first two components obtained from antioxidant, granulometry and colour data of pollen samples with the prevalence of species present both in unifloral [8] and actual samples. Eigenvalues, explained and cumulative variance and loadings of the variables are also reported.

**Figure 3 molecules-27-07996-f003:**
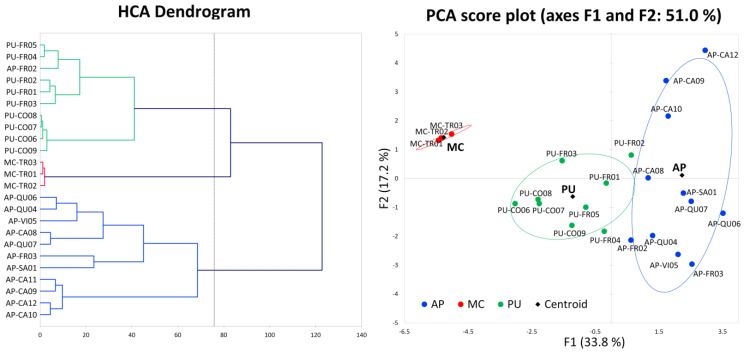
Multivariate statistical elaboration on the whole dataset incorporating 24 bee pollen samples and 22 variables (palynological antioxidant and granulometry data). HCA: dendrogram obtained by cluster analysis, using Ward algorithm and Euclidean distance to space the cluster. PCA: score plot of the first two components.

**Table 1 molecules-27-07996-t001:** Bee pollen samples. Zone and period of harvest, prevalent pollen and percentage, protein and moisture content (%) are described. Samples were labelled with two letters identifying the province, two letters identifying the prevalent botanical species and a consecutive number for each province.

Code	Period	Zone	Province	Protein	Moisture	Prevalent Pollen	
				%	%	Type	%
PU-FR01	22 April	Isola del Piano	Pesaro-Urbino (PU)	22.49	16	*Fraxinus ornus*	57
PU-FR02	26 April	Isola del Piano	Pesaro-Urbino (PU)	20.64	21.7	*Fraxinus ornus*	51
PU-FR03	29 April	Isola del Piano	Pesaro-Urbino (PU)	18.66	22.2	*Fraxinus ornus*	64
PU-FR04	3 May	Isola del Piano	Pesaro-Urbino (PU)	19.97	14.8	*Fraxinus ornus*	73
PU-FR05	6 May	Isola del Piano	Pesaro-Urbino (PU)	20.58	17.8	*Fraxinus ornus*	63
PU-CO06	28 June	Isola del Piano	Pesaro-Urbino (PU)	20.8	14	*Coriandrum*	63
PU-CO07	1 July	Isola del Piano	Pesaro-Urbino (PU)	20.52	13.5	*Coriandrum*	73
PU-CO08	3 July	Isola del Piano	Pesaro-Urbino (PU)	20.1	12.8	*Coriandrum*	76
PU-CO09	7 July	Isola del Piano	Pesaro-Urbino (PU)	19.88	15	*Coriandrum*	80
AP-SA01	21 April	Cavaceppo	Ascoli Piceno (AP)	21.09	15.8	*Salix*	40
AP-FR02	30 April	Cavaceppo	Ascoli Piceno (AP)	21.42	13.9	*Fraxinus ornus*	68
AP-FR03	8 May	Cavaceppo	Ascoli Piceno (AP)	24.52	11.9	*Fraxinus ornus*	44
AP-QU04	15 May	Cavaceppo	Ascoli Piceno (AP)	19.88	13.8	*Quercus ilex*	66
AP-VI05	22 May	Cavaceppo	Ascoli Piceno (AP)	20.84	13.1	*Vitis*	50
AP-QU06	29 May	Cavaceppo	Ascoli Piceno (AP)	18.04	14.9	*Quercus ilex*	76
AP-QU07	5 June	Cavaceppo	Ascoli Piceno (AP)	17.78	11.3	*Quercus ilex*	41
AP-CA08	12 June	Cavaceppo	Ascoli Piceno (AP)	16.02	16.9	*Castanea*	36
AP-CA09	19 June	Cavaceppo	Ascoli Piceno (AP)	18.81	12.9	*Castanea*	76
AP-CA10	26 June	Cavaceppo	Ascoli Piceno (AP)	20.75	10.5	*Castanea*	84
AP-CA11	3 July	Cavaceppo	Ascoli Piceno (AP)	22.1	10.8	*Castanea*	100
AP-CA12	9 July	Cavaceppo	Ascoli Piceno (AP)	21.97	13.2	*Castanea*	100
MC-TR01	7 July	Matelica	Macerata (MC)	21.93	14.9	*T. alexandrinum*	46
MC-TR02	9 July	Matelica	Macerata (MC)	21.54	16.5	*T. alexandrinum*	55
MC-TR03	12 July	Matelica	Macerata (MC)	22.24	14.7	*T. alexandrinum*	50

**Table 2 molecules-27-07996-t002:** Total phenolic content (TPC) and antioxidant activity data of studied bee pollen samples. Samples are grouped and also mediated by province. Superscript letters within each column indicate homogeneous subclasses resulting from Tukey’s post hoc multiple comparison test (*p* < 0.05) performed between all samples (uppercase) or between means of data of the same province (lowercase).

	TPC	ABTS	DPPH	ORAC
	mg GA/g DW *	μmol TE/g DW **	μmol TE/g DW **	μmol TE/g DW **
PU-FR01	18.3 ± 1.7 ^E^	180.7 ± 14.9 ^C^	50.7 ± 9.8 ^HIJK^	553.4 ± 79.8 ^EF^
PU-FR02	23 ± 2.2 ^D^	211.8 ± 20.7 ^BC^	69.9 ± 8.1 ^EF^	669.9 ± 79.2 ^DE^
PU-FR03	19.1 ± 1.2 ^EF^	160.7 ± 18.3 ^DE^	52.1 ± 11.2 ^HIJKL^	491 ± 68.9 ^G^
PU-FR04	16.1 ± 1.7 ^FG^	151.9 ± 12.8 ^D^	48.9 ± 8 ^HIJKL^	464.4 ± 66.3 ^FG^
PU-FR05	15.2 ± 1.3 ^GH^	148.5 ± 16.7 ^DEF^	52.7 ± 8.2 ^HIJK^	452.2 ± 40.7 ^GH^
PU-CO06	13.9 ± 1.1 ^HIJ^	123.9 ± 9.6 ^FG^	43.8 ± 5.2 ^JKL^	359.9 ± 42.5 ^HIJ^
PU-CO07	16.6 ± 1 ^EF^	139.9 ± 14.7 ^DEF^	40.9 ± 7.5 ^KL^	442.9 ± 59.7 ^G^
PU-CO08	16.2 ± 1.5 ^EF^	145.8 ± 18 ^DE^	40.7 ± 6.1 ^KL^	453.4 ± 40.1 ^FG^
PU-CO09	17 ± 2 ^EF^	143.4 ± 11.3 ^DEF^	43.8 ± 9.3 ^KL^	404.9 ± 35.1 ^GHI^
PU samples	17.3 ± 1.5 ^ab^	156.3 ± 15.2 ^b^	49.3 ± 1.22 ^a^	476.9 ± 56.9 ^b^
AP-SA01	20.8 ± 1.6 ^D^	195.2 ± 20.1 ^C^	54.2 ± 9.4 ^GHI^	759.7 ± 86.2 ^ABC^
AP-FR02	17 ± 1.4 ^EF^	177.5 ± 17.4 ^C^	39.4 ± 7.7 ^LM^	617.6 ± 62.3 ^DE^
AP-FR03	16.8 ± 1.5 ^EF^	187.5 ± 16.9 ^C^	30.3 ± 7.5 ^M^	701.9 ± 62.5 ^BC^
AP-QU04	17.5 ± 1.8 ^E^	182.8 ± 19.7 ^C^	47.3 ± 7 ^HIJKL^	559.8 ± 59.3 ^E^
AP-VI05	17.3 ± 1 ^E^	180.8 ± 18.5 ^C^	61 ± 5.9 ^EFG^	704.2 ± 77.9 ^BC^
AP-QU06	21.3 ± 1.9 ^D^	192.8 ± 19.2 ^C^	55.7 ± 10 ^FGH^	783.9 ± 47.1 ^AB^
AP-QU07	20.2 ± 2.7 ^D^	188.9 ± 17 ^BC^	79.1 ± 10.5 ^D^	653 ± 84.7 ^CD^
AP-CA08	17.2 ± 1.3 ^EF^	152.1 ± 14.1 ^DE^	71.9 ± 11.6 ^E^	475.6 ± 56.5 ^FG^
AP-CA09	23.9 ± 1.8 ^C^	192.6 ± 16.3 ^BC^	101.7 ± 8.5 ^C^	607.8 ± 54.8 ^DE^
AP-CA10	24.4 ± 1.7 ^C^	205 ± 14.3 ^B^	110.5 ± 6.7 ^B^	716.6 ± 61.8 ^ABC^
AP-CA11	28.5 ± 1.9 ^B^	233.8 ± 16.4 ^A^	143.2 ± 8 ^A^	772.5 ± 93.6 ^A^
AP-CA12	31 ± 2 ^A^	257.6 ± 15.6 ^A^	147.4 ± 9.9 ^A^	801.6 ± 117.5 ^A^
AP samples	21.3 ± 1.7 ^a^	195.5 ± 17.1 ^a^	78.4 ± 8.6 ^a^	679.5 ± 72.0 ^a^
MC-TR01	12.8 ± 0.9 ^IJ^	118.6 ± 18.2 ^G^	49.2 ± 8.9 ^HIJKL^	328.2 ± 35 ^IJ^
MC-TR02	12.8 ± 1.2 ^J^	112.1 ± 20.2 ^G^	45.7 ± 10 ^IJKL^	300.1 ± 44.3 ^J^
MC-TR03	14.1 ± 1.4 ^HI^	128.3 ± 18.5 ^EFG^	53.5 ± 9.1 ^GHIJ^	338.7 ± 62.6 ^IJ^
MC samples	13.2 ± 1.2 ^b^	119.7 ± 18.9 ^b^	49.5 ± 9.3 ^a^	322.3 ± 47.3 ^c^
Total samples	18.8 ± 1.6	171.3 ± 16.6	63.9 ± 8.5	558.9 ± 63.3

* mg gallic acid equivalents per g of dry pollen; ** μmol Trolox equivalents per g of dry pollen.

**Table 3 molecules-27-07996-t003:** Instrumental colour data (CIE L*a*b* colour space coordinates) of the surface of integer (I) and finely milled (M) loads and pollen load size distribution.

Code	L* (I)	a* (I)	b* (I)	L* (M)	a* (M)	b* (M)	>2400	2400–2000	2000–1690	<1690
PU-FR01	60.4	4.4	40.5	65.6	6.8	50.1	16%	46%	28%	11%
PU-FR02	58.2	7.6	41.3	63.4	8.4	48.1	17%	40%	29%	14%
PU-FR03	58.1	8.5	46.2	64.4	9.0	53.1	6%	43%	30%	21%
PU-FR04	60.7	6.2	46.9	64.3	9.2	53.6	18%	59%	16%	7%
PU-FR05	61.3	7.7	46.7	64.2	8.2	54.3	16%	49%	22%	12%
PU-CO06	50.5	8.0	33.0	57.7	11.2	44.8	7%	55%	27%	11%
PU-CO07	50.8	9.3	32.5	56.9	12.3	48.0	9%	55%	22%	14%
PU-CO08	51.8	9.2	35.1	57.8	11.3	47.8	8%	54%	27%	11%
PU-CO09	50.6	11.1	33.1	55.5	11.5	46.5	16%	59%	18%	7%
AP-SA01	57.4	6.9	37.6	62.5	7.5	51.8	34%	43%	13%	11%
AP-FR02	61.5	5.8	43.3	64.3	7.2	48.2	20%	56%	19%	4%
AP-FR03	57.7	4.8	35.0	62.3	5.1	41.6	41%	48%	8%	3%
AP-QU04	59.6	6.0	40.2	64.8	7.0	52.5	27%	58%	11%	5%
AP-VI05	60.5	6.0	47.3	62.7	7.4	53.9	27%	61%	11%	1%
AP-QU06	62.0	6.4	49.6	61.6	7.8	52.5	46%	45%	7%	2%
AP-QU07	60.4	8.2	51.9	61.0	11.0	61.9	36%	49%	10%	4%
AP-CA08	57.3	9.3	43.7	60.5	14.8	68.1	37%	42%	15%	6%
AP-CA09	60.8	6.9	41.8	64.0	10.2	61.0	29%	32%	26%	12%
AP-CA10	61.8	6.4	43.5	66.6	6.9	53.3	18%	50%	24%	8%
AP-CA11	65.1	6.0	52.6	68.3	7.1	58.8	39%	40%	18%	4%
AP-CA12	66.8	5.0	52.1	69.2	6.9	58.4	19%	41%	32%	8%
MC-TR01	47.4	8.4	31.0	53.2	11.1	47.6	8%	39%	31%	22%
MC-TR02	46.7	9.9	34.4	52.4	11.1	50.6	10%	38%	34%	18%
MC-TR03	49.7	8.3	31.4	53.7	12.5	51.3	8%	39%	31%	22%

## Data Availability

Not applicable.

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
