# Peer review of "Characterisation of Bee Pollen from the Marche Region (Italy) According to the Botanical and Geographical Origin with Analysis of Antioxidant Activity and Colour, Using a Chemometric Approach"

_molecules, 2022, doi:10.3390/molecules27227996_

Round 1

Reviewer 1 Report

The authors conducted interesting study on bee pollen. Studies of the amounts of phenolic compounds and their antioxidant activity generally seek to answer the question of how valuable the pollen of the corresponding composition are. So, the idea of research is relevant. In general, the research is described clearly, the results are presented and the data processed statistically.

The Title and keywords of the article are clear and inform readers about the main research directions.

Abstract. I would suggest that this section should be supplemented, as it emphasizes the applied methods and research results are not sufficiently presented.

The Introduction provides a detailed analysis of general trends in studies that have focused on the pollen nutritional and botanical properties.  In my opinion, the authors need to justify the innovativeness of their research in more detail in the introduction.

Results. The results of the research are analyzed with the justification of statistically significant differences, as many as 13 other authors are cited. Very strange, but the discussion section continues the analysis of the results and only 3 literature sources are cited. This is fundamentally inconsistent with the content of the sections of the article. In my opinion, it is simply possible to combine these two sections into one, Results and Discussion section.

Materials and methods. A review of Table 1 shows that very different pollen samples were compared. In some fields, only 36 to 80 percent of the pollen of the dominant plants were found, and in some there were detected pollen of one plant. The question arises, why are quantitatively different samples compared and how reliable are such results? The composition of the pollen sample depends exclusively on how the researchers chose the time to collect the material also.  Why were such terms of material collection chosen? Thus, the statement in the conclusions "Comparison of these results with those previously obtained from monofloral bee pollens show higher values for multifloral pollen suggesting that the floral species present even in low percentage may have an influence on the nutraceutical properties of these products" is just a hypothesis.

Reviewer 2 Report

In the study, pollen samples from an Italian region were analysed aiming a classification according to botanical and geographical characteristics that were also correlated with physicochemical and antioxidant properties. The study also considered changes in frequency of pollen types during the flowering seasons. In general, the manuscript is well conceived with a systematic study and appropriate data analysis. However, the differentiation between monofloral and multifloral is not discussed, so that the basis of the analysis is quite weak.

Specific comments:  

36: “... therefore the survival of the colony honey …” Are you sure it is the survival of honey?

68: “...On this basis, various multifloral pollens were collected in three different locations ...” Why are pollen samples considered multifloral, if the frequency of the dominant pollen is over 45% (see previous work [8]!). How is “multiforal” defined?

178 “... clarness ...” Consider revising.

325: “pollen samples were collected in 2015 in the Marche region, Central Italy, by...” How is the situation 7 years later?

244-250: “... Comparing the results obtained in the analysis of the antioxidant capacity of multifloral pollens reported in Table 2 with those previously obtained by our research group in the analysis of monofloral bee-pollens [8], it can be seen that the antioxidant activity values measured for multifloral pollens are generally higher than those of monofloral pollens. Furthermore, if data obtained from multifloral pollens with a prevalence of a specific species are compared with those of monofloral pollens and with those collected in different areas, the results are not entirely consistent.” Usually, a frequency of pollen over 45% is considered typical for monofloral. In the study, almost all samples could be classified as monofloral. How should this be understood? Definition of monofloral or multifloral has not been discussed in the manuscript.

250-252: “This suggests that even the floral species present in low percentage may have an influence on the nutraceutical properties of these products.” This affirmation needs a more profound analysis.

Reviewer 3 Report

Overall manuscript is written well. Few corrections can be performed to enhance the Manuscript and also improve the quality of Manuscript. Author prepare a good manuscript. Kindly rectify few of the errors listed below

Figure 1: between line 88-89: The species written in normal font they must be in italics and Umbelliferae Forma species must be start with small ‘f ’ as it is the species rest other are if genera pls specify it

Line 110: ABTS, DPPH and ORAC abbreviations are not with complete forms

Table 2 : Data abbreviations in table are not clear and detailed it in the legend. pls revise

Line 124 to 127: Not much meaning is coming so rewrite with scientific results (the text having considerable and could have words not relevant with specific results)

507 Line: The journal name in the reference is wrong it is Nutrients instead of Current status and therapeutic potential

Line 536:  Journal abbreviations should be with full stop pls rewrite Ital. J. Food Sci.

Line 559: Apis mellifera must be in italics

Discussion with more references to better explain it

Main conclusions should be part of abstract
